# If You Want to Prevent Hamstring Injuries in Soccer, Run Fast: A Narrative Review about Practical Considerations of Sprint Training

**DOI:** 10.3390/sports12050134

**Published:** 2024-05-15

**Authors:** Pedro Gómez-Piqueras, Pedro E. Alcaraz

**Affiliations:** 1Paris Saint Germain Soccer Club, Rue Guy Crescent, 78300 Poissy, France; 2UCAM Research Center for High Performance Sport, UCAM Universidad Católica de Murcia, Campus de los Jerónimos 135, 30107 Murcia, Spain; palcaraz@ucam.edu

**Keywords:** professional, football, sprinting, prevention, HSI

## Abstract

Hamstring strain injuries (HSIs) are one of the most common injuries in sprint-based sports. In soccer, the ability to sprint is key, not only because of its relation to performance but also due to its possible protective effect against HSIs. Although many authors have focused on the “how”, “when”, and “what” training load should be implemented, there is a lack of practical proposals for sprint training in a high-level professional environment. The objective of this narrative review is, after a deep review of the scientific literature, to present a practical approach for sprint training, trying to answer some of the questions that most strength and conditioning coaches ask themselves when including it in soccer. Once the literature published on this topic was reviewed and combined with the practical experience of the authors, it was concluded that sprint training in soccer, although it presents an obvious need, is not something about which there is methodological unanimity. However, following the practical recommendations from this narrative review, strength and conditioning coaches can have a reference model that serves as a starting point for optimal management of the internal and external training load when they wish to introduce sprint training in the competitive microcycle in professional soccer players, with the aim of reducing HSIs.

## 1. Introduction

Depending on the specific context of each sport, the objective of the athlete when performing a sprint is to cover a certain distance in the shortest possible time [1]. During a soccer game, a player covers an average of 17 to 81 sprints, with 2 to 4 s duration, and usually over distances of less than 20 m [2]. Approximately 70% of all these actions appear when the player is already moving and not when they are standing in a position [3]. Thus, the inclusion of training sprints in soccer must consider that these actions are mainly “flying sprints”, with a curvilinear rather than a linear trajectory [4].

All these high-speed efforts have increased in recent years in professional soccer [5]. In most cases, goals and decisive actions are preceded by such efforts [6]. Soccer players must be physically well prepared to deal with these demands [7]. A greater adaptation to sport demands and the use of multifactorial prevention protocols [8] could be the explanation for the apparent general reduction in injuries [9]. However, hamstring strain injuries (HSIs) are still the most common in soccer [10], and their incidence has not experienced a trend similar to that of other injuries, neither in games nor in training sessions. The UEFA data demonstrate that HSIs have increased by 4% annually in recent seasons [11] and now constitute 24% of all injuries in men’s professional soccer [12]. HSIs have a significant performance and economic burden [13], leading to an average of 90 days and 15 games lost per club per season [14]. The majority of HSIs occur when players run over 25 km/h and above 80% of their maximal speed [15], and present recurrence rates of 18% [12]. When this injury occurs during competition, it has recently been shown that it is preceded by a short period (5 min) of an unusual amount of running at more than 21 km/h [16].

Specifically, upon analysis of the most commonly injured muscles within the hamstring group, it appears that the long head of the biceps femoris (BFlh) is the most frequently damaged [17]. The complex and multifactorial nature of this injury [18] makes it difficult to attribute its appearance based on an isolated risk factor [19]. We know that, although in most cases it appears during high-speed efforts and sprints [15], there are several factors that could cause it. Some of these factors include: inadequate management of training loads [20], inadequate weekly exposure to high-speed running [21], the existence of previous injury [22], neuromuscular fatigue [23], low levels of force (isometric, concentric and eccentric) [24], poor lumbopelvic control [25] and improper sprint mechanics [26,27]. These are some of the main factors that could cause HSIs [28], but especially, as some authors point out, there is a lack of multicomponent programs, where all the aforementioned factors should be taken into consideration [8].

In sprinting, we are faced with a reality where, on the one hand, the player needs to be able to perform actions at a very high speed during matches, and on the other hand, this type of effort increase the risk of muscle injury. Although this paradox could lead us to see sprinting sometimes as a “friend” and sometimes as an “enemy”, there is currently a strong consensus on the need to include it systematically and regularly in our training schedule [29,30,31,32,33]. However, to the best of our knowledge, and as is supported by other authors [34], there is no clear agreement that describes how to include it, especially when considering all the training load variables (intensity, volume, programming, etc.) from an evidence-based approach.

Therefore, the objective of this study is to present an evidence-based practical proposal for sprint training in professional soccer teams. How can we introduce it in our training? When should we do it? How can we monitor and evaluate it? How much do we need it? These are some of the questions that we will try to answer.

## 2. Methods

In order to provide sufficient theoretical support for this practical proposal, a search strategy was designed to be able to reach all relevant materials published in databases. A systematic search was carried out during the months of May and June 2023. The Medline, Embase and Sportdiscus databases were explored without restrictions regarding the year of publication. The following keywords in English were used in the search strategy combined with different boolean markers: (sprint training OR speed training OR high intensity training) AND (football OR soccer) AND (professional OR semi-professional) AND/OR (injuries OR hamstring strain injuries OR hamstring injuries OR noncontact injuries). Additionally, the authors used select reference lists to identify relevant papers that were not found in the initial search. Studies were considered relevant if they recruited an athletic population: this includes individuals competing in any individual or team sport, or well-trained individuals that demonstrate a high level of conditioning. Because we chose to conduct a narrative review rather than a systematic review, we did not strictly follow the guidelines for Preferred Reporting Items for Systematic Reviews and Meta-Analyses (PRISMA). The initial search showed 486 results. After adding the inclusion criteria, the total number of texts was 197. Exactly 68 were included after reading the abstract. Finally, after reading the full text, 24 were included based on the quality of the studies. Final decisions for article inclusion were made by consensus of the authors of this study.

## 3. Findings and Practical Proposals

### 3.1. Sprinting as a Protective Factor

When a player sprints, the activation caused in the hamstrings (level and timing), is very different from the strengthening and prevention exercises typically performed for this muscle group [35]. When we compare the effects of sprinting with the most commonly used exercises for preventing hamstring injuries such as Nordic Hamstring exercises, it becomes evident that sprinting activates the long head of the biceps femoris (BF) more. In particular, research shows that the activation of the BF is higher during sprinting than in Nordic Hamstring exercises [31].

When a player exceeds 80% of his/her maximal sprinting speed (MSS), the activation of the BF is significantly greater than the activation experienced by the rest of the hamstring muscles involved [36]. Specifically, as running speed increased from 80% to 100%, the BFlh activity during the terminal swing phase increased by an average of 67%, while the semitendinosus and the semimembranosus only showed a 37% increase [37]. According to previous studies, the higher activation of the BF muscle could be a result of attempting to slow down the forward movement of the limb [38,39]. This phenomenon could also explain why the BF muscle is often injured during high-speed activities [37]. According to some authors, activating this muscle group reaching speeds higher than 90% MSS during training results in a lower risk of injury [21].

Only if the quality and technique of the movement are adequate [27], specific programming of high-speed training would have a protective effect as a “vaccine” against HSIs [29]. For example, it has recently been shown that an increase in anterior pelvic tilt results in a significant nonuniform increase in tissue elongation in all regions of the three hamstring muscles (greater in the proximal than in the distal region); hence, if we want to prevent injuries, we must know the impact of this when we demand high-speed efforts from the athlete [40].

According to certain authors, the protective effect of running may be associated with leg stiffness (K_leg_) [41]. K_leg_ has been proposed to be of substantial importance during sprinting, because it generates an increase in ground reaction forces that facilitate the storage and return of elastic energy, thus increasing the stride frequency and reducing ground contact time [42]. Lower K_leg_ has been linked to inefficient storage and, in running kinematics and movement efficiency, is associated with an increased load on the contractile muscle units, which, in theory, also increases the risk of HSIs [27]. The number of factors influencing mechanical stiffness during running (age, running technique, sporting background, fatigue, running distance, etc.) makes it difficult to formulate clear and general conclusions about training recommendations [43]. Nevertheless, it remains evident that in order to attain optimal stiffness, this “vaccine” must be tailored to the soccer context and adapted to the specific capabilities and running mechanics of the player [44]. Otherwise, positive and protective adaptations will not be elicited. In addition, it should be noted that the relationship between sprint training and injury appearance is U-shaped, thus an inappropriate dose, insufficient or excessive, could increase the risk of injury significantly as well [33] (Figure 1).

Although sprint training is considered an effective strategy to reduce the risk of injury, we cannot assume that we can avoid injuries with solely this type of work. As mentioned above, we know that HSIs are complex and multifactorial, so their prevention must be addressed through multicomponent programs that generate sufficient chronic adaptations from the preseason, where other multiple variables are included (load control, strength work, individual deficits, physiotherapy, etc.) in addition to sprint training [8].

### 3.2. Sprint Training Proposal for Soccer Players

The vast majority of the scientific literature supports the importance of exposing the athlete to high speeds during the training week [45,46]. This does not mean that the implementation of sprint training is without doubts and controversies. Many contextual variables are specific to each team and they will determine the way sprint training is applied. For example, not all methodologies support a similar periodization of content throughout the week [47,48]. It is known that for tactical periodization, there is one day a week (Match Day-2 = MD-2) dedicated to stimulating speed [49]. Other methods of training consider that high speed will be achieved sufficiently working in wide spaces during MD-4 and or MD-3 days [7]. The experience of the players and the confidence of the technical and medical staff with sprint training, the competitive density of the team, and the differentiation of work between starters and non-starter players are some of the variables that must be taken into account before adjusting the weekly sprint dose. However, today, there are still doubts about how to train the sprint, when to place it in the training week, and how much is necessary.

#### 3.2.1. Determine How You Will Integrate It within Your Particular Training Environment

A combination of different methods (from the most analytical to the most specific of the sport) seems to be the most effective strategy to optimize sprint ability in our players [50,51]. Regardless of the chosen method, the need to approach sprint training in a multidirectional way has recently been proposed [52]. The model proposes training for acceleration, deceleration, change in direction, curvilinear sprint and sprint in a straight line, as these constitute key factors for the optimal development of sprinting. These components should be included throughout the training week, if our aim is to develop robust and fast soccer players in all directions (360°).

Considering all the manifestations proposed by the multidirectional model of speed, the inclusion of maximal sprint is usually the most difficult in some contexts and with some players. It is possible that the weekly team dynamic or the preferences of the head coach do not allow for the specific incorporation of sprint training. Additionally, a player’s injury history or beliefs could hinder the inclusion of these types of exercises. Although it is important to have a special sensitivity in determining the needs of our players, this does not mean that we should reject sprint training because most players have never trained in this way. Therefore, it is critical to understand how to adapt it to our reality and appropriately introduce this particular type of training.

As it is likely that you will find a wide variety of these considerations and particularities within your team, the safest approach is to introduce the sprint in a very progressive manner from the preseason [2]: avoid spikes in volume and intensity, which could cause injury [53] and allow players to progressively adapt to sprint training [54]. To progressively include these situations, we will use the “control-chaos continuum” proposed by some authors for the rehabilitation of injured soccer players [55]. In this approach, the prescribed tasks accumulate and increase in complexity based on the chaotic elements that compose them. In this way, we will be able to periodize the sprint training sessions throughout the different weeks of training, taking into account the complexity and specificity of the tasks. Initially, we will use tasks of high control and low complexity, where the number of stimuli is low. The individual analytical work (in a straight line or in a curve) is a very simple and pragmatic way to calibrate the dose and limit unwanted speed peaks during the first sessions. A collective finishing drill without opposition, over a maximum distance of 40 m, could be a good way to progress in specificity and complexity.

As a player adapts to sprint training and increases the speed values, we can add chaos to the tasks and even combine different methods. We believe that if we expose the player to real game situations where they have to sprint, the protective effect will be greater since the coordination demands required will be more similar. In this sense, our training sessions should focus on the context of the game and not just on the frequency and duration of bouts the player performs [56]. In the current conceptualization of high-intensity activity training, the “why” is equal to or more important than the “how much” [46]. If we understand the physical–tactical context in which sprints occur, drill design can better reflect these physical demands using an integrative approach [57,58].

If we want to maintain consistency in this contextual specificity, simulating game situations where “attack versus defense transitions” are prescribed could be one of the most appropriate approaches [59]. The range of stimuli that can be introduced and modified in these situations is vast [2]: from duels with a reduced number of players, to more complex situations with a greater number of participants; from situations with a numerical advantage for the attackers, to totally opposite situations. The starting area on the field, the positional disposition of the players, the relative distances between them, the start from standing or moving, the recovery time between actions and or their density, etc., are just some of the variables that we can modify. Figure 2 shows a proposal for weekly periodization of multidirectional sprint training based on its complexity.

Of all the possibilities, it is important to know which players are mostly exposed to sprinting and the different neuromuscular demands that each modification entails. For example, if we focus on the collective, the speed peaks reached by the defenders will likely be greater if we give a space–time advantage to the attackers [60]. If we focus on the individual, we should be aware that sprinting from standing does not require the same demands as flying sprints [61]. In addition, we should consider that the neuromuscular demands of the inner leg are not the same as the outer leg during a curvilinear sprint [4].

#### 3.2.2. Choose the Right Moment

Regardless of the chosen methodology, the training goal is to find the correct loading dose that generates positive adaptations without increasing the risk of injury [62]. To accomplish this, and although there are different ways of programming weekly training content, as long as the recovery from the previous match has been sufficient, the load should be greater on the days furthest from the next game (MD-5, MD-4 and MD-3), and it should decrease as we get closer to the next competition [7,63]. During the last two days of the microcycle (MD-2 and MD-1), we usually observe a tapering phase where reducing neuromuscular fatigue and getting ready to compete are prioritized [48,64].

Some professional clubs, over the last few years, have chosen the “Tactical Periodization” framework as a guide to distribute the different physical qualities during the week. Although this methodology has been applied in soccer for more than 20 years, it should be noted that science has not examined it in depth [65]. Explained in a very simplistic and reductionist way, this methodology allows us to dedicate one day of the week to each of the basic physical qualities: MD-4 for strength, MD-3 for endurance and MD-2 for speed. It has recently been shown that, if applied correctly, this way of distributing loads is optimal and does not generate inappropriate neuromuscular fatigue [66].

Performing speed training on the MD-2 day is a controversial issue for which there are no scientific studies to help defend our position. Although a positive association has been found recently between training speed on the MD-2 day and a lower incidence of injury [67], some professionals consider that the accumulated load during the MD-4 and MD-3 sessions is sufficient, and exposing the player to speed work with this level of fatigue may be counterproductive [63]. From our viewpoint, there are two key elements when it comes to justifying this training on MD-2: firstly, the volume and density of the efforts proposed, and, secondly, and perhaps more importantly, the assimilation capacity of our players to these stimuli (closely related to their experience in this weekly dynamic).

Regardless of the periodization model used, our opinion is that we must be convinced of the importance of stimulating the sprint during the week, regardless of the selected day. In microcycles where a single match is scheduled, our way of proceeding is to analyze the high-speed efforts made on MD-4 and MD-3 so that, together with the individual needs of each player, we can adapt the dose of speed to be applied on MD-2. Of course, if the player presents high levels of fatigue due to the accumulation of neuromuscular load from previous days, perhaps maximum sprints should be avoided or reduced on MD-2. This neuromuscular fatigue could be assessed with some isometric testing protocols in the posterior chain, week by week in MD+1 (to control the neuromuscular fatigue after the match) or MD-2 (if you want to test the neuromuscular readiness of the player after MD-4 and MD-3 sessions). This protocol is highly sensitive to neuromuscular fatigue compared to other ways of measuring lower body neuromuscular fatigue (i.e., knee maximal voluntary isometric contraction), or internal load (i.e., RPE) after sessions with different volumes of sprints, accelerations and decelerations [68].

When the competitive density is relatively higher, when more than one game per week is identified, the logic to include sprint training is equal: analyze the high-speed volume already performed (either in training or in competition) to be able to adapt its future dose. Finding moments to stimulate this ability can be difficult, especially for players who participate in one of the matches and need to dedicate more time to recovery. In this regard, we should remember that the recovery time after a speed session is 48–72 h [69], but this time will depend on the dose received. Thus, another strategy we use when completing speed training on MD-2 is to limit the number of decelerations by reducing braking and sudden changes in direction. It has been shown that this type of effort, due to its greater muscular demand in an eccentric regime, generates greater neuromuscular fatigue [70].

Finally, we point out the importance of sprint training in players with less playing time. There is a significant difference between the weekly external load demands for players who tend to start (starter) and those who participate less in the competition (non-starters) [71]. For this group, we must find “extra” moments where we can complete high-speed training and thus compensate for their needs [72]. Depending on multiple variables, this training can be added immediately after matches or on subsequent days. The key point, as noted above, is to ensure that all players receive an optimal dose of weekly speed, not too high and not too low [73]. Figure 3 presents some ideas for scheduling this type of training during the week.

#### 3.2.3. Sprint Monitoring

Before addressing the question that every reader asks, “*How much sprint do my soccer players need?*”, it is convenient to clarify certain peculiarities that could occur when monitoring this variable. It is generally consented by the scientific community that determining the activity profile of a soccer player with GPS requires different categorized speed zones ranging from 0 to 36 km/h. The lack of a consensus to determine the thresholds for each of these categories can lead us to fall into biased decisions and premature conclusions when analyzing our data and comparing them with others [74]. Most GPS devices use either 24 km/h or 25 km/h as the cut-off point to classify an effort as a sprint [75]. However, it is not appropriate to use the same speed threshold for all players while measuring sprints. This is because individual differences between athletes must be taken into account [75].

Using an absolute threshold (e.g., 24 km/h) when quantifying the number of sprints of two players with different peaks of speed (MSS) (Player 1: 33.6 km/h; Player 2: 29.7 km/h) could lead us to overestimate the values of one and underestimate the other. For Player 1, reaching 24 km/h would mean achieving 71.4% of their MSS, while for Player 2, it would be 80.8%. Since the effort required by Player 1 to reach the sprint is less than the effort required by Player 2, comparing the values of both players using absolute values would not be appropriate in all situations. Figure 4 shows a comparison of two players, who during a session performed the same number of sprints using absolute values (>24 km/h) but have very different relative numbers (>85% MSS).

Complementing the information provided by the absolute thresholds with that provided by the relative thresholds could be an interesting strategy [76]. In our case, in addition to the use of an absolute threshold common for all (24 km/h), we categorized the number of sprints for each player into 2 levels: sprints above 85% of the MSS and sprints that exceed 95% of MSS. Some authors have found that these relative thresholds show a greater association with the risk of suffering a non-contact injury [77]. To determine these thresholds, as suggested by some authors, we tend to measure the MSS through analytical tests, since these provide higher values than the values obtained during a competitive match [78].

#### 3.2.4. Searching for the Right Dose

First of all, there is no ideal number of sprints to which we should expose the player weekly. There is no magic dose for all players that would allow them to perform optimally without being injured. Each player’s position, their behavior in competition and their training dynamics are some of the variables that can bring us closer to this adequate dose. Based on this knowledge about the player, we recommend establishing an individual sprint profile specific to our context, for the training week as well as for competition. More specifically, we need to determine the amount of sprint training required by the competition as well as our training methods for each player. Only in this way will we be able to determine the approximate dose to which we must expose our players, so that they are optimally prepared during matches [21]. This question is important because, recently, it has been shown that exposure to high speed and sprint during training could be insufficient if we compare it with exposure in competition [79].

In delineating our sprint dosage, we typically adhere to a benchmark where each player, throughout the week leading up to match day (MD-4, MD-3, MD-2 and MD-1), experiences a high-speed workload akin to that encountered during actual gameplay. While contentious discussions persist around this dosage regimen, several studies suggest “training/competition ratios” falling within the range of 0.5 and 1.3 approximately [48,80,81]. Though not prescriptive data, this range serves as a preliminary guiding framework for our training approach.

In any case and for this purpose, the competitive profile by player position will be of great help (Figure 5). The data of this profile, being obtained through a multi-camera system installed in the stadiums, are not interchangeable with the data provided by the individual GPS devices [82]. To limit this problem, some authors have proposed the use of mathematical calibration equations that allow us to exchange the data provided by different measurement systems [83].

This profile must be complemented with the one obtained by our GPS devices during the training week. In this way, we will obtain a clearer overview of the approximate values in which each of our players must be located (Figure 6). It should also be noted that, during our weekly microcycle, we try to ensure that of all the sprints covered and measured in an absolute way, 35–40% of their total volume should exceed 85% of the MSS and 15–20% exceed 95%. Hence, we ensure that all players have been exposed to high-demand match situations. In line with this idea, the use of different recovery durations between sprint efforts, sometimes reduced, allows us to anticipate difficult scenarios that could occur in competition [84], such as in the initial and final minutes of each match, where the density of high-speed efforts is apparently greater [85]

Furthermore, if we examine the collective dose and its weekly periodization, we can observe that our sprint values are significantly higher on MD-2 than the rest of the days (Figure 7). Logically, this higher demand on this capacity must be compensated with a lower demand on other load variables. In this way, we will not lose the positive effects of pre-match tapering [86]. Finally, we want to underline, once again, that these values and references are specific to our context (methodology, type of player, game model, etc.), and trying to replicate them exactly in a different context could mean an inappropriate dose with unwanted consequences.

#### 3.2.5. Measure Whenever You Can, but Measure Properly

The final recommendation is that testing and measuring should be incorporated whenever possible and feasible. Data in sports are a complementary source of information that can make our training process much more efficient if used properly. Collecting and accumulating data and not making decisions based on them is a waste of time. If we measure retrospectively, we must try to know why it happened and how this information can serve us in the future [87].

In our case, being aware that the sprint can have risky and adverse effects if we do not use it in its proper measure, we usually quantify our sprint values “*in real time*” using the antennas that most GPS companies provide us, especially during MD-3 (endurance work in large spaces) and MD-2 (speed) training, as they are the ones in which the sprint can appear more frequently. In this manner, we can meticulously adjust the load during the training session. At the same time, we can adapt the individual dose of players according to their weekly needs. Keep in mind that if you do not make certain adjustments during the session, it is likely that you will not be able to apply them in subsequent training.

The assessment of the isometric strength of the posterior chain as a marker of neuromuscular fatigue is another strategy that can help us when adjusting the load of each player. A loss of strength or increased asymmetry between both legs are markers of acute fatigue that may be related to the risk of suffering HSIs [88]. For this reason, measuring this level of strength in the posterior chain could inform us of a previous load that is poorly tolerated by the athlete, in order to adjust their future load [89]. The McCall supine test, in its two variants (90° and 30°) [90], together with a more specific standing modification (IPF 90°−20°) [91], is one of the most interesting assessments in this sense. In addition to this isometric component, it could be useful to measure the eccentric response of the hamstrings with devices such as Nordbord^®^ (Vald Performance, Australia) [92], or the explosive component of the posterior chain with the recently proposed “*Ballistic Hip Thrust Test*” [93].

Finally, knowing that several investigations provide empirical data to support the association between sprint mechanics and HSI occurrence, a qualitative screening tool has currently been proposed that offers a practical approach to in-field movement assessment [94]. The S-MAS is a 12-item qualitative movement screening tool assessing the overall movement quality of an individual’s sprint running mechanics (trailing limb extension, back kick, trunk and pelvic rotation, thigh separation, anterior pelvic tilt, etc.), which could be useful to us. Scores are summed, with a total score of 0 indicating optimal mechanics and 12 suboptimal, with higher scores generally representative of poorer technique.

## 4. Practical Applications

Considering that sprinting is a key action in soccer and using it too little or too much can cause injuries, specific work must be carried out taking into account the demands of the game and the team. Below are some summarized recommendations based on our professional experience (Figure 8):-Combine different training methods (from more analytical to more specific) throughout your macrocycles, mesocycles and microcycles.-Include flying sprints from different positions with multidirectional stimuli (acceleration, deceleration, change in direction, curvilinear sprint and sprint in a straight line) when working on speed. For example, 10-20 m for acceleration, or 45 m flying sprints (15 + 30 m) for maximum velocity should be ideal.-Progress in complexity as your players adapt to different speed stimuli and game situations applying the “control-chaos continuum”.-Distribute the weekly load appropriately, avoiding excessive neuromuscular overload and favoring the necessary supercompensation processes in the days before the game.-Create your own sprint profile for competition and training based on your particular context and remember that no “magic” dose is valid for everyone. As suggestion, “training/competition ratios” falling within the range of 0.5 to 1.3 approximately, can be used. During the microcycle, 35–40% of their total volume should exceed 85% of the MSS, and 15–20% should exceed 95%.-Define the sprint properly. Remember that using similar absolute thresholds for all players is not the best way to individualize their needs.-Evaluate whenever you can, but only if the collected data will be useful for making future decisions. The isometric strength of the posterior chain could be a helpful strategy when adjusting the load of each player. The S-MAS is a qualitative screening tool which allows us to measure the quality of the athlete’s movement during the sprint.

## 5. Conclusions

Although there is sufficient theoretical evidence to support the need and importance of introducing sprint training in soccer, we can verify that there is no methodological unanimity from a practical point of view. We believe that the proposal presented here, although it has been developed in a specific context, can be useful as a justified theoretical/practical model that serves as a starting point for those professionals who are considering the introduction of this type of training to reduce HSIs.

## Figures and Tables

**Figure 1 sports-12-00134-f001:**
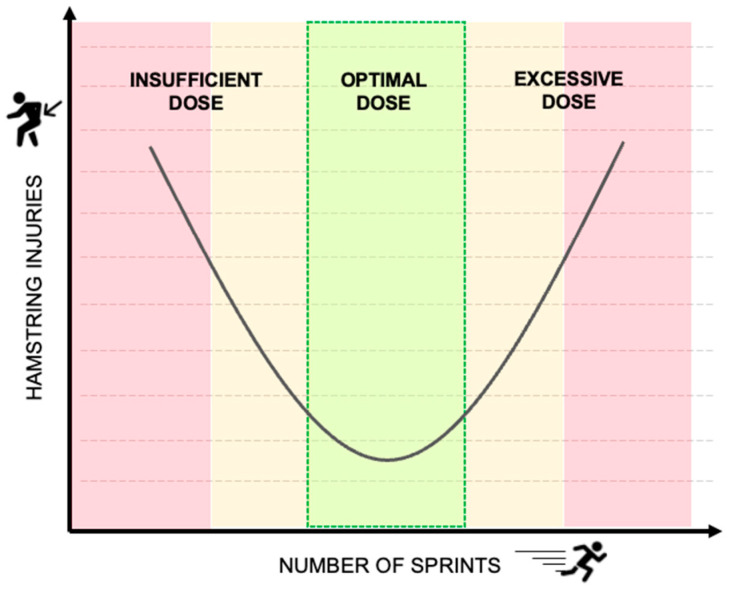
Relationship between the number of hamstring injuries and sprint dose.

**Figure 2 sports-12-00134-f002:**
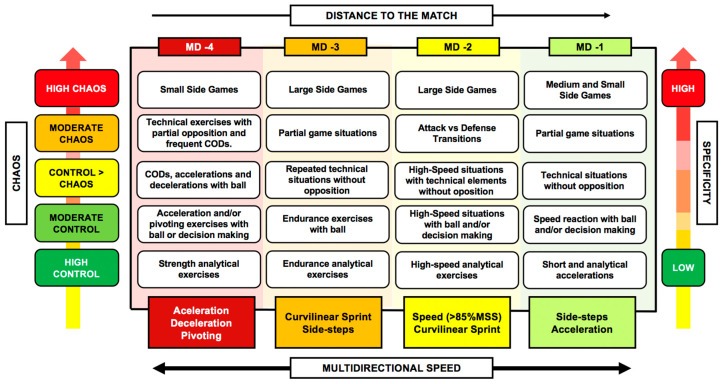
Proposal for the inclusion of the multidirectional speed model during the microcycle based on the complexity of the tasks (MD: matchday; CODs: changes of direction; MSS: maximal sprinting speed).

**Figure 3 sports-12-00134-f003:**
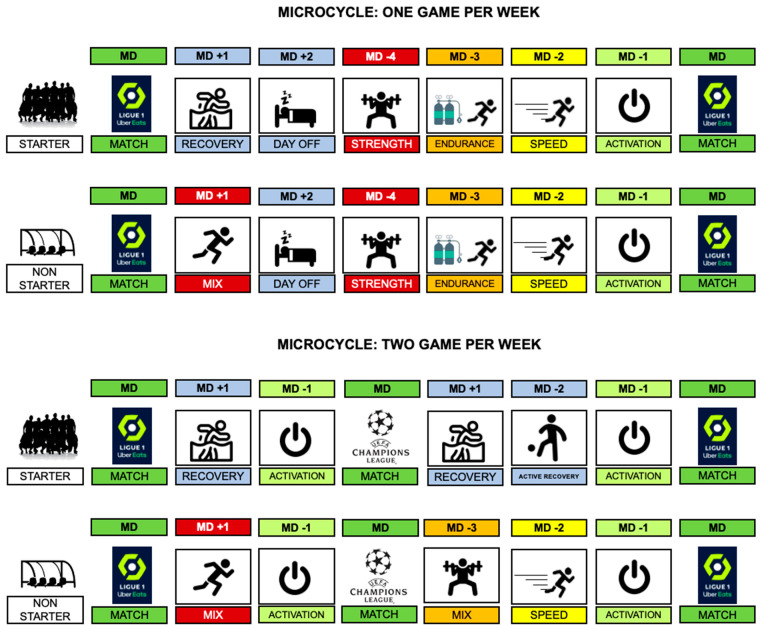
Proposal for weekly periodization in different microcycles.

**Figure 4 sports-12-00134-f004:**
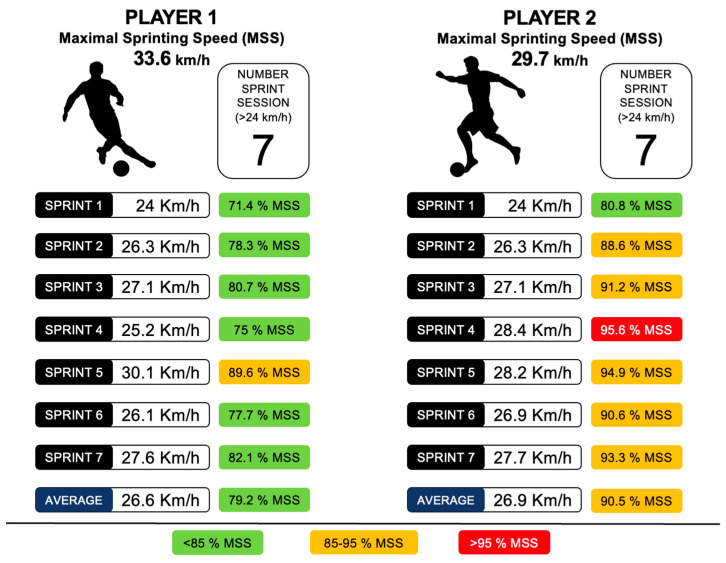
Sprint data comparison with absolute thresholds vs. relative thresholds.

**Figure 5 sports-12-00134-f005:**
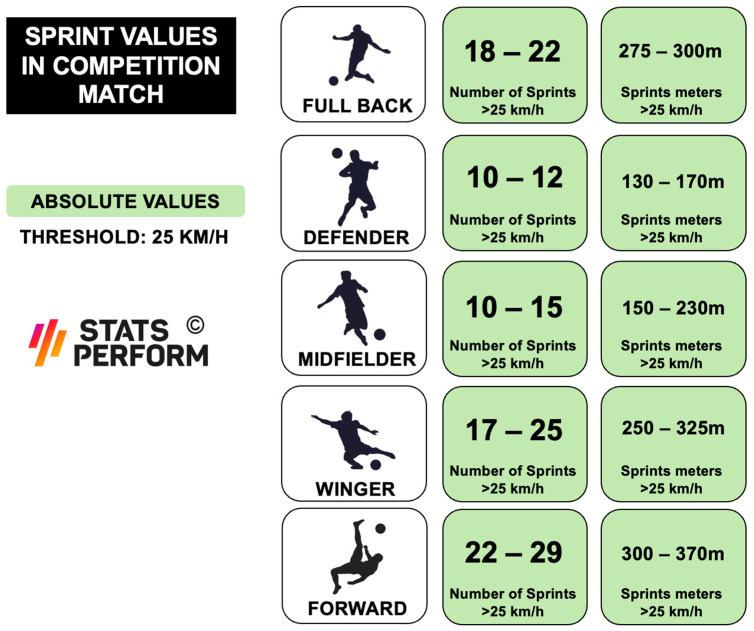
Absolute values by player position in competition (stats perform: multi-camera system in French league).

**Figure 6 sports-12-00134-f006:**
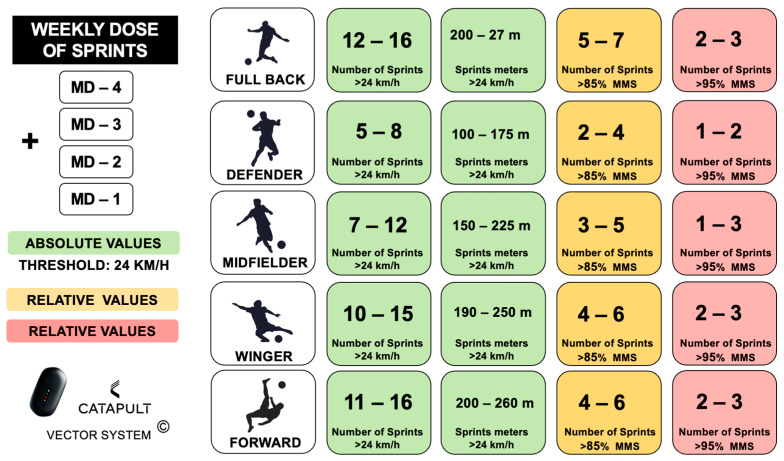
Absolute and relative values by player position in training (GPS-Catapult System).

**Figure 7 sports-12-00134-f007:**
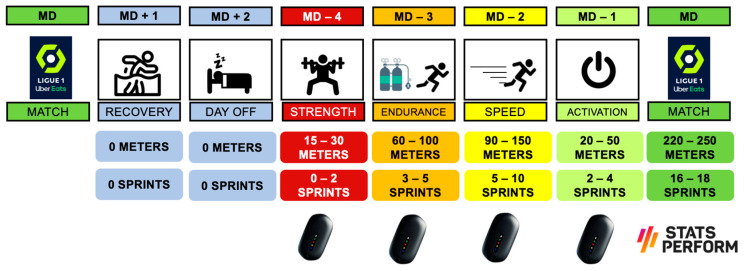
Sprint periodization during a typical competitive microcycle.

**Figure 8 sports-12-00134-f008:**
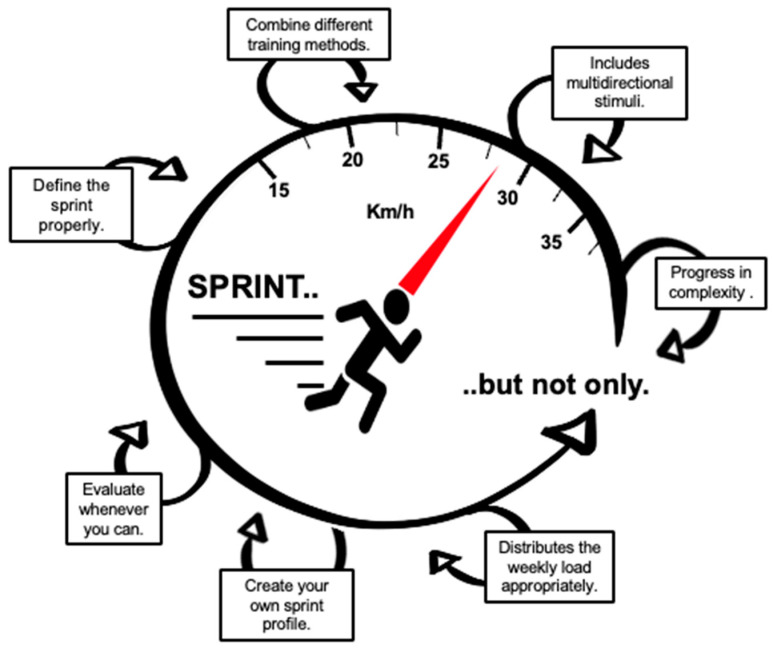
Evidence based practical applications.

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
