# Peer review of "If You Want to Prevent Hamstring Injuries in Soccer, Run Fast: A Narrative Review about Practical Considerations of Sprint Training"

_sports, 2024, doi:10.3390/sports12050134_

Round 1

Reviewer 1 Report

Comments and Suggestions for Authors

Review / If you want to prevent hamstring injuries in soccer, run fast: A practical approach to sprint training

Congratulations on a very well-written evidence-based practical article. Although the majority of the things should be familiar to the S&C specialist, you summarized and properly collected them. However, I have some minor suggestions to improve your manuscript.

Introduction:

When this injury occurs during competition, it has recently been shown that it is preceded by a short period (5 min) of unusual running at more than 21 km/h – What do you mean by unusual? Unusual high amount of running at these velocities or something else?

Sprint training proposal for soccer players:       

To accomplish this, and although there are different ways of programming weekly training content, the load should be greater on the days furthest from the next game (MD-5, MD- 4 and MD-3) – MD-5 is usually MD+2 (or similar) and that is for sure not proper moment to make sprint sessions. Please amend this part as we need to take into consideration the recovery miny block from the last game.

Explained in a very simplistic and reductionist way, this methodology allows us to dedicate one day of the week to each of the basic physical qualities. The MD-4 for strength, the MD-3 for endurance and the MD- 2 for speed. – Although the concept of tactical periodization indeed is as stated, I personally don't believe that it is possible to target only one ability during sessions as they are always multidirectional in terms of adaptations.

This neuromuscular fatigue could be assessed with some isometric testing protocols in the posterior chain, week by week in MD+1 (to control the neuromuscular fatigue after the match) or MD-2 (if you want to test the neuromuscular readiness of the player after MD-4 and MD-3 sessions) – How exactly does this protocol look like?

Please amend figure 6. as it can not be seen fully.

You are saying that players need 48-72 hours to recover from speed training but still perform speed sessions on MD-2? Please explain.

Practical applications:

Very well written!

Author Response

Review / If you want to prevent hamstring injuries in soccer, run fast: A practical approach to sprint training

Congratulations on a very well-written evidence-based practical article. Although the majority of the things should be familiar to the S&C specialist, you summarized and properly collected them. However, I have some minor suggestions to improve your manuscript. 

Thank you very much for your evaluation and good reception of our work.

We agree with you that some of the things raised here should be perfectly known by any SC coach.

However, since it is a current topic, we believe that compiling some of them in the same document would be very helpful for those professionals who work in sports training.

Introduction:

When this injury occurs during competition, it has recently been shown that it is preceded by a short period (5 min) of unusual running at more than 21 km/h – What do you mean by unusual? Unusual high amount of running at these velocities or something else?

Yes. The authors refer to an unusual distance at these speeds. Following your recommendation and to avoid confusion, we have added the word "amount" to clarify it.

Sprint training proposal for soccer players:        

To accomplish this, and although there are different ways of programming weekly training content, the load should be greater on the days furthest from the next game (MD-5, MD- 4 and MD-3) – MD-5 is usually MD+2 (or similar) and that is for sure not proper moment to make sprint sessions. Please amend this part as we need to take into consideration the recovery miny block from the last game.

According to our experience and according to the reading of the cited work, MD-5 and MD+2 do not have to be the same day.

Your team can play on a Friday and then play on Sunday of the next week. That would give us a long microcycle of more than 7 days.

For this reason, what the authors want to explain with this statement is that the greater the distance from the next game (and as long as the recovery from the previous game has been complete), the load could be higher (not just talking about the sprint).

We have added “ as long as the recovery from the previous match has been sufficient” for clarify that

Explained in a very simplistic and reductionist way, this methodology allows us to dedicate one day of the week to each of the basic physical qualities. The MD-4 for strength, the MD-3 for endurance and the MD- 2 for speed. – Although the concept of tactical periodization indeed is as stated, I personally don't believe that it is possible to target only one ability during sessions as they are always multidirectional in terms of adaptations.

Totally agree with your opinion. If we delve deeper into this methodology, we can see that prioritizing the development of a structure does not mean reducing the complexity of human movement and this sport.

For this reason, at the beginning of the paragraph we indicate that it is a simplistic and reductionist explanation, but sufficient from our point of view for the objective of our article.

This neuromuscular fatigue could be assessed with some isometric testing protocols in the posterior chain, week by week in MD+1 (to control the neuromuscular fatigue after the match) or MD-2 (if you want to test the neuromuscular readiness of the player after MD-4 and MD-3 sessions) – How exactly does this protocol look like?

In section 2.5 we introduce a paragraph that includes the most appropriate tests from our experience to evaluate this neuromuscular fatigue. We believe that it is a sufficient explanation for this article and that the reader can delve deeper into it by going to the original references.

Please amend figure 6. as it can not be seen fully. 

Corrected.

It was a formatting error that occurred when transferring our document to the template provided by the journal.

You are saying that players need 48-72 hours to recover from speed training but still perform speed sessions on MD-2? Please explain.

Traditionally it has been noted in all publications that sprint work requires a recovery time that varies from 24 to 72 hours. Our logic tells us that this time will depend on the dose applied (this is what we indicate with our phrase "but that this time will depend on the dose received").

This means that a soccer player can be exposed to a small dose of sprinting (for example 80-100 meters) without this being a problem 48 hours later. This is what our experience tells us in recent years but there are no scientific articles that corroborate it. It would therefore be an interesting line of research since not all sprint doses require the same recovery time.

Practical applications: 

Very well written!

Reviewer 2 Report

Comments and Suggestions for Authors

Dear authors,

Thank you for the opportunity to review this paper. I congratulate the authors on this work, which was clearly deep. The topic is interesting, and the purpose of this work will contribute to several professionals involved in soccer. However, I must encourage the authors to revise their manuscript presentation (i.e., during the text are missing spaces between words; numbered subsections are duplicated; figures are cut; use of a more “scientific writing”; etc.). I also have great concerns regarding the type of manuscript you presented. This seems a literature review, but no methods are described. Based on this, I have made some comments below, but I am not able to accept this manuscript in its current form.

Abstract

The abstract is exclusively focused on the manuscript’s purposes. I recommend the authors include details regarding the methods used, the results achieved, and the main conclusions. 

Introduction

At the end of the Introduction, the authors mention load variables (intensity, volume, etc.), which seem connected to the questions defined in the last paragraph. However, in the text, the authors describe sprint ability and HSIs. I believe that adding information regarding the methods typically used to promote sprint training would be useful to better contextualize the reader about the topic that is approached in this study.

Methods

I suppose that this paper is a literature review. Please add details regarding the methods used during the data search, databases, keywords, inclusion/exclusion criteria, etc.

Note that subsections are wrongly numbered. For instance, there are several 2.1. subsections.

References

References do not follow the journal’s guidelines and some are missing or do not exist in the reference list (i.e., reference 97).

Author Response

Dear authors,

Thank you for the opportunity to review this paper. I congratulate the authors on this work, which was clearly deep. The topic is interesting, and the purpose of this work will contribute to several professionals involved in soccer. However, I must encourage the authors to revise their manuscript presentation (i.e., during the text are missing spaces between words; numbered subsections are duplicated; figures are cut; use of a more “scientific writing”; etc.). I also have great concerns regarding the type of manuscript you presented. This seems a literature review, but no methods are described. Based on this, I have made some comments below, but I am not able to accept this manuscript in its current form.

First of all, we would like to thank you for the time you have spent reviewing our article.

We are pleased that you consider it interesting and we believe that thanks to your indications its quality will increase significantly.

Specifically, we would like to point out that we have corrected all the formatting errors that you point out to us.The problem occurred when our original document was transferred to the journal official template by some of its l designers. For this reason these types of errors existed.

In general, it is true that from a scientific point of view it is difficult to place our article in any of the sections usually used. It is not a systematic review, nor does it pretend to be. For this reason, a methodological protocol such as the PRISMA rules has not been followed.

The objective of our article is to create "practical" science for everyday life, relying on more theoretical science. For this reason we combine the literature already published (which has been thoroughly reviewed) with our practical experience over the last few years with high-level soccer teams. We believe that this mix, although it is difficult to fit within a usual section of any journal, will be of great interest to sports training professionals.

Abstract

The abstract is exclusively focused on the manuscript’s purposes. I recommend the authors include details regarding the methods used, the results achieved, and the main conclusions. 

Following your recommendation, we have added the main conclusion of this work at the end of the abstract.

Introduction

At the end of the Introduction, the authors mention load variables (intensity, volume, etc.), which seem connected to the questions defined in the last paragraph. However, in the text, the authors describe sprint ability and HSIs. I believe that adding information regarding the methods typically used to promote sprint training would be useful to better contextualize the reader about the topic that is approached in this study.

We consider that most ways to carry out this type of training are included throughout the text. If you can tell us anything more specific that you consider important to add, we will be happy to do so to improve our article.

Methods

I suppose that this paper is a literature review. Please add details regarding the methods used during the data search, databases, keywords, inclusion/exclusion criteria, etc.

As we have tried to explain, our article is not a review per se since methodologically it does not follow the necessary steps for it. In any case, we have added a paragraph where we explain the way in which we have accessed the information that supports our practical proposal presented here.

Note that subsections are wrongly numbered. For instance, there are several 2.1. subsections.

Corrected. Error when transferring the document to the template

References

References do not follow the journal’s guidelines and some are missing or do not exist in the reference list (i.e., reference 97).

Corrected. Error when transferring the document to the template

Reviewer 3 Report

Comments and Suggestions for Authors

Pedro Gómez-Piqueras  and Pedro E. Alcaraz wrote the Atricol “If you want to prevent hamstring injuries in soccer, run fast: A practical approach to sprint training”. Autors discuss the role of sprint training in preventing hamstring injuries in professional soccer players. It highlights the importance of sprinting as a protective factor, as it activates the hamstrings differently from traditional exercises and can reduce injury risk when performed correctly.

Even if it is well written there are some concerns about the way the article was prepared.

I recommend the authors to use the PRISMA methodology to determine the accuracy of the data collection and how are processed.

https://www.prisma-statement.org/

The authors should formulate discussions and conclusions, because the article does not have an ending, and discussions and conclusions are missing.

Author Response

Thanks for you time and dedication.

We clearly agree with you.

 Our article does not follow a methodological sequence typical of what would be a systematic review.

For this reason, this type of guides like the one you indicate (PRISMA), designed for carrying out systematic reviews, have not been followed.

Our work, which is difficult to place in a regular journal section, is a practical proposal based and supported by published literature and the authors' experience over years with professional soccer players.

Although it does not follow a strictly scientific pattern from a methodological point of view, the journal encouraged us to present it since its practical value for the reader was considered interesting.

In any case, and following your recommendations, we have added an introductory paragraph explaining how to extract the theoretical information on which our practical proposal is based. Also following your advice, we have added a conclusions section.

We believe that adding more sections, just to be consistent with a PRISMA methodology, would not be correct, especially because they are steps that have not been followed as strictly as PRISMA recommends.

Round 2

Reviewer 2 Report

Comments and Suggestions for Authors

Dear authors, I appreciate your effort in improving this manuscript. I understand your point of view regarding the manuscript type. You have mentioned that the literature was reviewed based on the authors' experience. However, how can the reader understand that the information included was the most relevant considering your topic? A description of the keywords used during the search was provided, even though how many papers resulted from your search, and how many did you use for the analysis? What was the time frame used? How did you select the papers of interest?

Maybe this paper can be described as a narrative review? If you agree, please update the title. 

Author Response

Thank you once again for your dedication and your wise recommendations.
Following the same, we have changed the title of the article (adapt it to a narrative review).
We have also made some adjustments to the abstract, search strategy and conclusions, so that these sections will better adapt to the new title.
We hope that now our work is at the level of this journal. Thank you

Reviewer 3 Report

Comments and Suggestions for Authors

The authors argued why they did not use PRISMA and made the necessary efforts to modify the article for publication.

Author Response

(The authors gave the same response as above.)
